# Development and Evaluation of Machine Learning-Based High-Cost Prediction Model Using Health Check-Up Data by the National Health Insurance Service of Korea

**DOI:** 10.3390/ijerph192013672

**Published:** 2022-10-21

**Authors:** Yeongah Choi, Jiho An, Seiyoung Ryu, Jaekyeong Kim

**Affiliations:** 1Department of Big Data Analytics, Kyung Hee University, 26, Kyungheedae-ro, Dongdaemun-gu, Seoul 02447, Korea; 2School of Management, Kyung Hee University, 26, Kyungheedae-ro, Dongdaemun-gu, Seoul 02447, Korea

**Keywords:** Korea NHIS, health checkup cohort DB, medical cost prediction, data imbalance, machine learning, logistic regression, random forest, XGBoost

## Abstract

In this study, socioeconomic, medical treatment, and health check-up data from 2010 to 2017 of the National Health Insurance Service (NHIS) of Korea were analyzed. This year’s socioeconomic, treatment, and health check-up data are used to develop a predictive model for high medical expenses in the next year. The characteristic of this study is to derive important variables related to the high cost of domestic medical expenses users by using data on health check-up items conducted by the country. In this study, we tried to classify data and evaluate its performance using classification supervised learning algorithms for high-cost medical expense prediction. Supervised learning for predicting high-cost medical expenses was performed using the logistic regression model, random forest, and XGBoost, which have been known to result the best performance and explanatory power among the machine learning algorithms used in previous studies. Our experimental results show that the XGBoost model had the best performance with 77.1% accuracy. The contribution of this study is to identify the variables that affect the prediction of high-cost medical expenses by analyzing the medical bills using the health check-up variables and the Korea Classification Disease (KCD) large group as input variables. Through this study, it was confirmed that musculoskeletal disorders (M) and respiratory diseases (J), which are the most frequently treated diseases, as important KCD disease groups for high-cost prediction in Korea, affect the future high cost prediction. In addition, it was confirmed that malignant neoplasia diseases (C) with high medical cost per treatment are a group of diseases related to high future medical cost prediction. Unlike previous studies, it is the result of analyzing all disease data, so it is expected that the study will be more meaningful when compared with the results of other national health check-up data.

## 1. Introduction

According to research related to high-cost medical expenses, it was found that the top 5 or 10% of medical expenses accounted for more than 50% of the total medical expenses distribution [1,2]. For handling increasing medical expenses and individual healthy life, Korean government have operated the health insurance system as a social security system. Therefore, Korea’s health insurance system started to be implemented for workplaces with 500 or more workers in 1977 and it has been expanded to nationwide medical insurance from 1989. After the Korean government announced the ‘Government 3.0′ policy in 2012, it began to open public data from 2013. Since opening, the number of public big data downloads as of 2017 has increased by about 170 times compared to that of 2013 [3]. Among Korea’s representative public big data in the medical field, the Health Insurance Corporation health check-up cohort Data Base (DB) consists of the data on the health check-up questionnaire and the history of medical expenses used by health check-up examinees [4].

The Korean government is making various efforts to reduce medical expenses that increase every year. Furthermore, to strengthen health insurance coverage and to maintain policy continuity, it is important to identify factors that increase medical expenses and make policy efforts to reduce social costs caused by diseases. In particular, since more than 50% of the total medical expenses distribution comes from the top 5–10% medical expenses, it is expected to be of great help in reducing medical expenses if high-cost medical expenses can be predicted by analyzing the results of annual health check-up. Therefore, this study derives factors influencing high-cost items for sustainable health care coverage using health check-up item variables.

Existing research related to high-cost prediction has predicted medical expenses for a specific disease, or used data accumulated over a short period of about 1–3 years as an input variable to learn a predictive model. In this study, not limited to high-cost prediction or disease morbidity prediction targeting specific disease groups, but reflected the entire 21 Korea Classification Disease (KCD) classification groups representing domestic disease groups in input variables to predict which disease groups are related to high cost. All the 21 major KCD groups representing all diseases in Korea are reflected in the input variables. The Korean medical claims data accumulated over many years were trained and used to predict which disease group is related to high cost medical expenses.

In this study, big data from the National Health Insurance Corporation representing the Korean population was provided and used for analysis to ensure the representativeness of the data. We defined high cost in predicting medical expenses as the top 10% of expenses for evert year. In the National Health Insurance Corporation DB, there are four types of variables, which are Socioeconomic variables, Health check-up item variables (laboratory values, personal and family history, lifestyle), Medical cost variables (costs used for main diagnosis and treatment in the past year), and Number of diagnoses and CCI Create analysis data by inputting correction scores (variables other than cost).

Supervised learning for predicting high-cost medical expenses was performed using the logistic regression model, random forest, and XGBoost, which have been known to result the best performance and explanatory power among the machine learning algorithms used in previous studies [5]. We derived the important variables that have a significant impact on high-cost prediction through the derived prediction model.

As a result of the experiment, a variable with high importance influencing high cost was age in the social qualification category. In the health check-up category, hemoglobin, BMI, and lipidemia (total cholesterol level, low-density cholesterol level) were found. In the cost-related category, it is whether the total cost of expenses used by one person is high. Lastly, a variable with high importance in the non-cost category was the number of major diagnostic visits in the year. Additionally, Musculoskeletal diseases (M) and respiratory diseases (J), which are the most treated diseases among the KCD disease groups, were identified as disease groups affecting the prediction of high medical expenses. Malignant neoplasm (C), which has a high medical cost per treatment, was also identified as a disease group related to the prediction of future high medical expenses. Our experimental results show that the XGBoost model had the best performance with 77.1% accuracy.

## 2. Related Work

### 2.1. High-Cost Medical Expenses Prediction

Table 1 summarizes existing studies focusing on methodology, prediction targets, and input variables. Based on these existing studies, this study uses Logit regression and Random Forest, which are known to perform well in previous studies, and uses XG boost that have excellent predictive performance and can select important variables among re-cent machine learning techniques. In addition, the high-cost patient criteria were defined as the top 10%. In addition, it was intended to include as many input variables as possible in previous studies.

Lee et al. [6] made cost predictions through comparison of classification regression tree performance with ANN with 492 patient data. They demonstrated the superiority of ANN’s performance in cost prediction using Demographic, Diagnosis, Number of treatment tests, Length of stay, Number of surgeries as predictors. Powers et al. [7] evaluated regression statistical modeling methods for predicting expected total annual health costs using the Pharmacy Claim-Based Risk Index (PHD). As a result of the study, it was confirmed that the pharmacy claim data PHD can be used to predict future medical costs. Koing et al. [8] analyzed the effect of complex injuries in which three or more chronic disease states occur simultaneously on medical costs. Conditional inference tree algorithms were used for 1050 people. Studies have shown that Parkinson’s disease and heart failure are the most influential predictors of total cost. Bertsimas et al. [9] analyzed cost prediction for the first time using supervised learning as a previous study that predicted future costs as a cost variable. For two years from August 2004 to July 2006, total insurance expenditure medical expenditure was predicted using regression decision trees and clustering methods using medical, demographic, and cost-related input variables. The results confirmed that using 22 cost-related variables and using the CART regression decision tree algorithm as a classifier showed the same performance as the analysis performance by adding medical and demographic variables (approximately 1500 variables). Sushmita et al. [10] evaluated the use of regression trees, M5 model trees, and random forests for cost prediction, and confirmed that M5 performed best. The prediction results confirmed that the previous medical cost alone could be a good indicator of future medical costs. To predict patients’ costs for the next year, they studied using a set of Medical Expenditure Panel Survey (MEPS) data taken from responses to panel surveys provided to households, employers, health care providers and insurance providers over a two-year period. Duncan et al. [11] compared several machine learning and statistical models to predict patient costs, including M5, Lasso, and boosted trees. We experimented with 30,000 patients using data from 2008 for training and using the total allowance of claims for 2009 for testing. Various predictors were used as input variables, including the total cost of the previous year, total medical expenses, total pharmacy costs, demographic information, total visits, and chronic diseases (83 different states). The results showed that boosted stress and M5 were the most effective classification methods in the *R^2^* and Mean Absolute Error (MAE) evaluation indicators, respectively, and that the cost predictor was a strong predictor. Itsuki et al. [13] used MinaCare data to perform a 2016 high-cost medical cost prediction based on data for two years from 2013 to 2015. The study was conducted based on a previous study (Park et al. [12]) that the high-cost prediction performance was improved when health examination data were used together in the previous study. The study found that the model for predicting medical costs using clinical data and billing cost data showed improved performance for high-cost patient prediction compared to the prediction model that uses only existing billing data.

According to previous studies related to medical cost prediction, in response to the question of the medical cost prediction approach reported in the existing literature, a research approach that uses historical expenses variables for the purpose of predicting future medical expenses is better than the research approach that uses clinical data [14]. Therefore, we found that it is important to integrate clinical data such as laboratory test results with cost variables for the high performance of high-cost predictions if possible.

### 2.2. Korea Public Health Information Data

As digital techniques develop, medical-related big data such as electronic medical record data, medical image data, genome analysis data, health check-up data, and disease are accumulating in the health care domain. A representative type of health and medical big data provided by Korean public institutions is the ‘Korean National Health and Nutrition Examination Survey’ big data provided by the Korea Centers for Disease Control and Prevention. Since the enactment of the National Health Promotion Act in 1995, it has been implemented since 1998 to understand the health and nutritional status of the people. The field of research consists of examination survey, health survey, and nutrition survey [15].

The second is big data held by the Health Insurance Review and Assessment Service. It has almost similar data to the National Health Insurance Corporation. Customized data is provided from 2007, and data on general specifications, medical treatment, illness and disease, outpatient prescriptions, and the current status of nursing homes are provided. The difference from the National Health Insurance Corporation big data is that it provides additional drug-related information [16].

The last one is the National Health Insurance Corporation big data. National Health Insurance collects and manages data on qualifications and insurance premiums from birth to death, hospital and hospital usage history, national health check-up results, rare incurability and cancer registration information, and medical benefit data [16]. The National Health Insurance Corporation builds a national health information database (national health information database) with accumulated information on subscriber qualifications and insurance premiums, health check-up history, medical treatment details submitted by medical providers, and medical institution information, and provides them for research purposes [17].

The National Health Insurance Corporation provides a sample research DB by sampling some of the national health information DB. There are sample cohort DB 2.0, health check-up cohort DB, elderly cohort DB, female cohort DB at work, and infant screening cohort DB as detailed types. According to the health check-up cohort DB, about 510,000 people were subject to general health check-ups between the ages of 40 and 79 between 2002 and 2003 among those who maintained qualifications in 2002 [18].

In this study, big data from the National Health Insurance Corporation, which represents the domestic population, was provided and used for analysis to secure representation of the data. We defined high costs as the top 10% of the year’s expenditures, and developed a machine cost model to explain whether to predict high costs using ma-chine cost models by using socioeconomic variables, health examination variables (lab figures and past history self-questioning, lifestyle (drinking, smoking, exercise frequency), medical cost variables, diagnostic frequency, and CCI correction scores (non-cost variables) in the DB.

## 3. Experiments

### 3.1. Dataset

To develop a high-cost medical expenses prediction model proposed in this study, the National Health Insurance Corporation sample health check-up cohort DB was provided and used for analysis. Figure 1 shows the overall process of extracting data needed in this study from the entire data. The unit n in Figure 1 represents the number of NHIS subscribers.

In this study, we collected data after receiving the approval of the IRB deliberation exemption from the Bioethics Review Committee of Kyunghee University (KHSIRB-21-385) and reviewing the data provision of the National Health Insurance Corporation’s health insurance data sharing service. After that, we merged three data tables into one table: a qualification table, a medical statement table, and a health check-up table among the health check-up cohort DB. As for the health check-up table, the main check-up and questionnaire items were changed due to the reorganization of the check-up system in 2009, and the pulmonary tuberculosis item was newly added from 2010, this study limits the subjects to those with health check-up data from 2010 to 2017. In the qualification DB table, we selected subjects who maintained health insurance qualifications during the study period by setting the year from 2010 to 2017, and the number of subjects was 3,785,617. Here, we selected 1,578,098 people who were eligible for health insurance from 2010 to 2017 and who had a history of health check-up from the health check-up DB table, excluding 2,207,519 people without a history of health check-up.

After that, from the treatment DB table, the state, injured disease, first hospitalization date, and cardiac care benefit cost were extracted from the table of medical health institutions, and from 2010 to 2018 for everyone with a health check-up history from 2010 to 2017. Derived variables such as the number of times of treatment, the amount of medical expenses by treatment group for the main and injured patients, whether the total medical expenses for the year were high, and the CCI disease correction score were created, and cost-related variables were merged into the qualification and health check-up extraction data. The Table 2 below shows the number of people and the standard amount of high-cost medical expenses for the composition of the dataset by year.

Table 2 shows the number of people and the standard amount of high-cost medical expenses for the composition of the dataset by year. The unit of the reference amount is expressed in Korean currency value people. High-cost medical expenses account for about 50% of the total annual medical expenses, and it was described that predicting and managing high-cost medical expenses is important in policy areas such as reducing medical expenses [1].

In this study, the top 10% of annual medical expenses were labeled as high-cost medical expenses and that might cause a class imbalance problem [12,13]. It is known that the class imbalance affects the performance in binary classification problems [19]. To solve that problem, various data sampling methods have been proposed to improve the performance of classification models in data imbalance conditions [20,21]. In this study, we used both under-sampling that reduces the number of class data with a large number of samples and over-sampling method that restores and extracts the minority category data and matches the ratio with the multi-category data to solve the class imbalance problem. In this study, we used the ROSE (Random Over Sampling Examples) method as over-sampling method, which alleviates the data imbalance problem by generating new data based on the down-sampling method and smooth bootstrap [22,23], before applying machine learning method.

The ROSE technique is a technique that repeatedly synthesizes and newly synthesizes data to be used for learning and is known to help improve classification prediction accuracy while avoiding the overfitting problem [24]. Additionally, that technique provides a unified framework that simultaneously solves the model estimation and accuracy problems of data imbalance learning, and alleviates the data imbalance problem by generating new data based on the so-called smooth bootstrap [22,25]. Table 3 summarizes the variable statistics of the training and test sets used in the study.

Using extant literature on predict future high-cost user analysis as a reference [12,13], we divided variables into four categories: Social qualification, health check-up, cost, non-cost. Table 4 summarizes the description and characteristics of all the variable used in this study.

### 3.2. Methods

High-cost medical expense classification prediction supervised learning was performed using the logistic regression [26,27] model, random forest [28], and XGBoost [29,30,31] method, which has been known to result the best performance and explanatory power among the machine learning algorithms used in previous studies that performed high-cost prediction research. In this study, data pre-processing and machine learning were conducted by remotely accessing the virtual room of the National Health Insurance Corporation. R studio version 3.3.3 was used as the analysis program used in the study. Figure 2 shows our experiment framework.

There are 12 training datasets used for machine learning. Logistic regression, random forest, and XGBoost algorithm were used as machine learning algorithms for 12 datasets that had completed the preprocessing process. By learning each of the three types of machine learning algorithms, the total number of machine learning was performed 36 times. A schematic diagram of the detailed dataset used for machine learning training is shown in Figure 3 below.

In this study, we learned a high-cost patient prediction model by adding variables hierarchically by referring to some previous studies [12]. By 12 training datasets, which were sequentially accumulated and combined 4 data types and 85 input variables, it was predicted whether the case of test data would have high medical expenses. As for the classification evaluation criteria, based on the predicted value of 0.5, 0.5 or less was classified as low cost (0), and 0.5 or more was classified as high cost (1).

## 4. Results

### 4.1. Unbalanced Data Preprocessing and Learning Results

Unbalanced data performed high accuracy by classifying classes from minority categories into majority categories for high accuracy, confirming that accurate machine learning training was not performed. The box-plot in Figure 4 is a diagram that visualizes the classification performance of the test set with an algorithm that trained the prediction model with the raw dataset 4. The average of the classes classified into the low-cost group (0) and the high-cost group (1) is shown, and there is no difference in the average between the two groups. The standard prediction value for distinguishing the high-cost group from the low-cost group is 0.5. In Figure 4, the predicted average value of the high-cost group is 0.2, and it can be seen that most of the actual high-cost groups are classified into the low-cost group.

On the other hand, it was confirmed that the accuracy of the machine learning result was improved when machine learning was performed after preprocessing the target ratio of the training data to 1:1. Figure 5 shows the average of the predicted values of the classes classified into the low-cost group (0) and the high-cost group (1). The distribution mean value is about 0.38, and it can be seen that the distribution mean of the predicted data of class 1 corresponding to high cost is 0.6 or more. It was confirmed that the machine learning results were improved compared to the raw data based on the discrimination criterion of 0.5.

### 4.2. Comparison of Model Performance

#### 4.2.1. Confusion Matrix (F1-Score)

F1-score calculated through the confusion matrix to confirm the best performing model among all experimental models is a harmonic average of precision and recall and is an evaluation index mainly used when data imbalance between classification classes is severe [32]. Table 5 shows results of prediction model performance evaluation of the entire experimental dataset. Among the experimental results, the logistic regression model up-sampling the data showed the highest performance with an F1-score value of 0.332.

#### 4.2.2. AUC

ROC curve and AUC can obtain the performance of the model for multiple threshold settings in classification problems [33]. The ROC curve shows the TPR (True Positive Rate) and FPR (False Positive Rate) graphs for different thresholds. *X*-axis (FPR) of the ROC curve means the ratio of predicting the actual normal data as abnormal data, and the *Y*-axis (TPR) represents to the probability of correctly predicting an abnormal data as abnormal data. For AUC performance evaluation, sub-optimal less than 0.5 worst, less than 0.7, 0.7–0.8 good, and excellent more than 0.8 are presented as performance evaluation criteria. By down-sampling, the ratio of target variables was preprocessed at a 1:1 ratio for each class, and the machine learning algorithm confirmed that the prediction model using the XGBoost method had an AUC value of 0.771 (77.1%), indicating that it was the best performing prediction model.

In the performance evaluation in Table 5 below, the Down-XGBoost performance index of dataset 1, which consists only of social qualification variables, was 65.3%. The AUC value improved by 3.9% (0.039) in dataset 2 with the added health checkup variable. When the cost variable was added in the dataset 2, performance index is 69.2% (social qualification variable + health checkup variable), and the AUC value improved by 7.7% (0.077). Finally, the dataset 3 performance index is 76.9% (social qualification variable + health checkup variable + medical care). In the case of the dataset 4, XGBoost model, which combines all variables including non-cost variables, the number of visits and the CCI correction score, the AUC value was improved by 0.2% (0.002), and finally the AUC performance of 77.1% was confirmed.

The Down-XGBoost model, which had the best performance evaluation to see how much the predictive power differs by variable type, was evaluated for each variable by varying the combination of each variable characteristic. Table 6 below shows the performance for each combination of Down-XGBoost variable categories. As a result, it was confirmed that the predictive model in which the cost variable (the cost variable used for the main diagnosis and illness group) was added to the basic social qualification variables resulted the best performance. In previous studies, it has been argued that the model performance is the most effective in predicting future costs only with cost variables [11]. In this study as well, when cost variables were added to the predictive model, the AUC value increased by 7.7%, confirming that the cost variables were highly important variables in predicting future medical expenses.

While the performance of the prediction model with social qualification and cost variables was 76.4%, the predictive model added non-cost variables such as the number of visits and CCI correction score results increased performance by 0.7% (77.1%). It was confirmed that predictive performance slightly increased when variables and non-cost variables were added.

#### 4.2.3. Variable Importance

As a result of machine learning, the top 30 variables of the predictive model that showed the highest performance were identified. The prediction model performance of dataset 4 learned from down-sampling training data was the highest, and the top 30 variables of the dataset 4 random forest model and the XGBoost model are shown in Table 7 below along with the variable characteristics.

## 5. Conclusions

In the experimental results of prediction models, it was confirmed that age (AGE) is the only variable that affects high-cost prediction among the characteristics of eligible variables as a demographic variable. In the Random Forest model, Mean Decrease Accuracy, the contribution to prediction accuracy, was the highest, and in the XGBoost model, the age variable had the highest importance in the top three. Accordingly, it was confirmed that the old is more likely to be classified as having high medical expenses.

It was confirmed that the health check-up item variables were hemoglobin, body mass index, total cholesterol level related to lipidemia, and low-density cholesterol level, which were found to be high-importance variables in both predictive models. In terms of screening variables, it was confirmed that body index and laboratory variable items that can be analyzed in blood and body fluid were important for prediction, and that self-questionary questions related to history and lifestyle were relatively insignificant in predicting high cost.

As for the variable with high importance in the predictive model in the cost variable, it was confirmed that the variable with the most influence was whether or not medical expenses were high in the previous year. Lastly, it was confirmed that common variables with high importance in predicting high cost among non-cost variables were the number of major diagnoses in the year immediately preceding the forecast year, the number of treatments in poor condition, and the CCI correction score, which is the risk index for comorbidities, as important variables in the predictive model.

## 6. Discussion

In this study, we developed a model to predict high-cost spenders to identify factors that increase medical expenses and help policy efforts to reduce social costs due to dis-eases, and to investigate future medical expenses increase factors by identifying important variables in the model algorithm. To conduct the research, we tried to predict the medical expenses related to the domestic medical expenses users by applying the variables of the health check-up items conducted by the state using a large sample of the health check-up cohort DB of the Korea Health Insurance Corporation. The implication of this study is that the empirical study was conducted by analyzing more than one million actual medical claim data and clinical data using public health and medical big data provided by the Korean government.

The implication of this study is that the empirical study was conducted by analyzing more than one million actual medical claim data and clinical data using public health and medical big data provided by the Korean government. Through the study, it was confirmed that musculoskeletal disease (M) and respiratory system disease (J), which are the most frequently treated diseases as major disease groups for high cost prediction of major disease groups in KCD, are disease groups that affect high cost prediction in the future. It was confirmed that cancer diseases are a group of diseases related to the prediction of high future medical expenses. Furthermore, a variable with high importance influencing high cost was age in the social qualification category. In the health check-up category, hemoglobin, BMI, and lipidemia (total cholesterol level, low-density cholesterol level) were found. In the cost-related category, it is whether the total cost of expenses used by one person is high. Lastly, a variable with high importance in the non-cost category was the number of major diagnostic visits in the year.

The limitation of this study is that, due to the nature of the data of the National Health Insurance Corporation, the cost variable is limited to the medical benefit claim data, so the data is composed based on the claim data of medical institutions. Because the training data and the target variable, the high cost of medical expenses variable, are based on the claim data, the data on non-reimbursable items were not reflected, so the items of medical expenses that were not covered by medical insurance were not learned.

Therefore, there is a limit to applying this research model when trying to predict whether high medical expenses will be actually spent, and there is a limit to predicting detailed disease names by grouping only KCD major disease groups and learning variables based on 21 major classification criteria.

In the future, by supplementing the limitations of this study, we intend to utilize the national medical information big data by using the KCD intermediate classification variable. The purpose of this study is to derive specific high-cost disease names by further subdividing the high-cost KCD-related disease groups derived from previous studies. When a high-level predictive model is developed using the health checkup DB variable, it is a promising research field to determine whether the predictive power of the health checkup variable is improved in the high medical cost predictive model.

## Figures and Tables

**Figure 1 ijerph-19-13672-f001:**
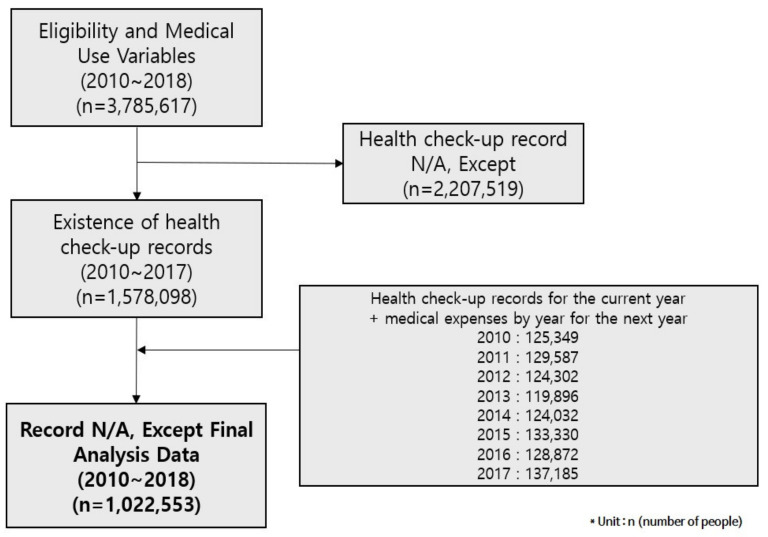
Generating data from the Korea National Health Insurance Corporation cohort DB.

**Figure 2 ijerph-19-13672-f002:**
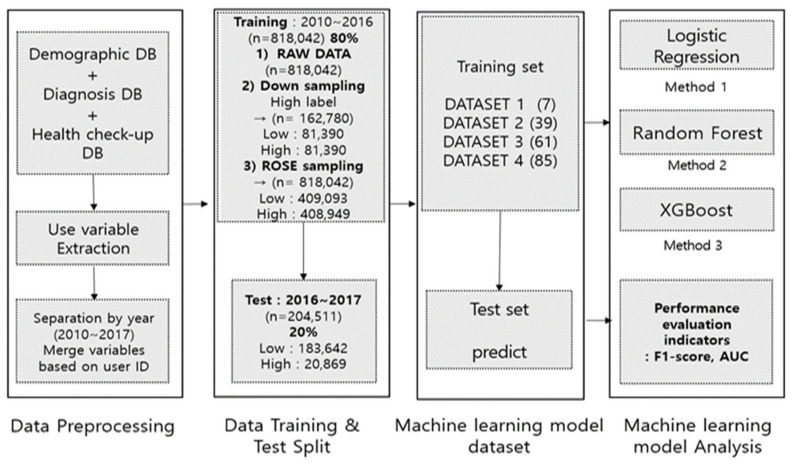
Experiment Framework.

**Figure 3 ijerph-19-13672-f003:**
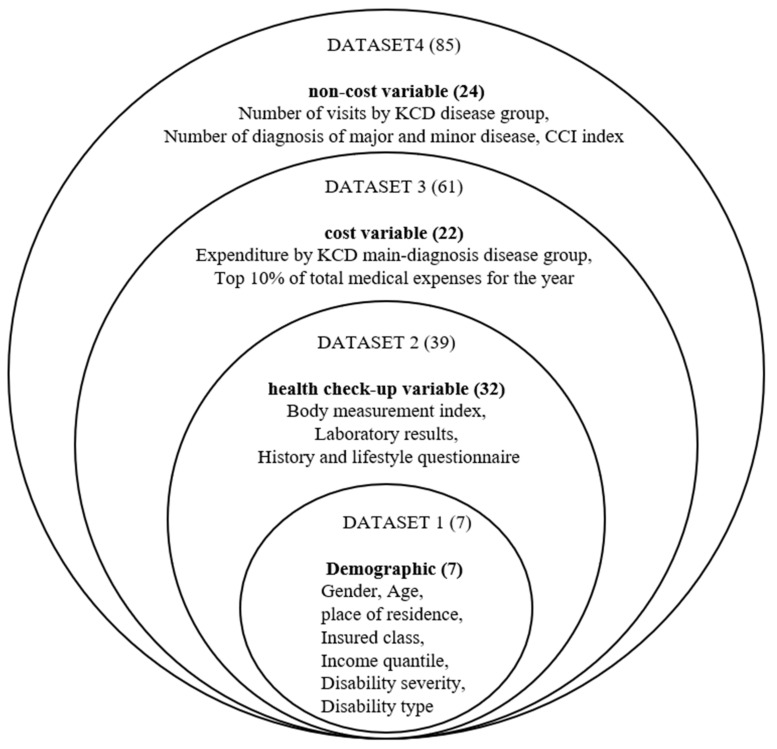
Variable diagram of dataset.

**Figure 4 ijerph-19-13672-f004:**
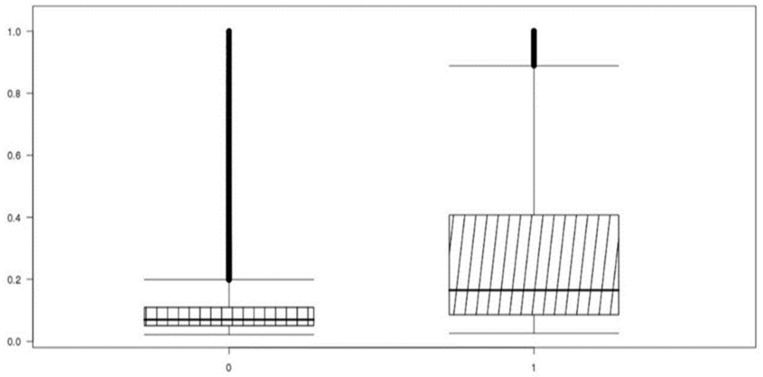
Raw dataset 4 (logistic regression) actual-prediction data distribution box-plot.

**Figure 5 ijerph-19-13672-f005:**
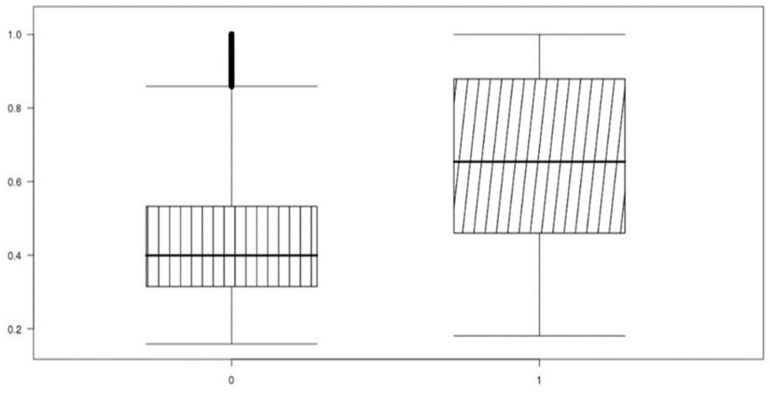
Down-sampling dataset 4 (logistic regression) actual-prediction data distribution box-plot.

**Table 1 ijerph-19-13672-t001:** Prediction of medical expenses performed prior research.

Researcher	Method	Target	Independent Variable	Metrics
Lee et al. (2004) [6]	Decision tree CART ANN	Total cost	Demographic Diagnosis Number of treatment tests Length of stay Number of surgeries, etc.	MAE
Powers et al. (2005) [7]	OLS Logistic regression GLM regression	Multi-class (bottom, top10%, top1%)	Demographic Pharmacy health Dimensions (PHD), etc.	Adjusted R^2^, MAE specificity PPV
Koing et al. (2013) [8]	CART CTREE	Total cost	Demographic Multimorbidity Drug dosage BMI, etc.	MAE
Bertsimas et al. (2008) [9]	Classification tree Clustering	Multi-class (five class)	Demographic Diagnosis Previous cost variables Total medical use cost, etc.	Hit ratio R^2^ MAE
Sushmita et al. (2015) [10]	M5 Random forest CART	Total cost	Demographic Comorbidity Index Diagnosis Medical use cost, etc.	MAE RMSE
Duncan et al. (2016) [11]	Gradient Boosting Decision tree Lasso regression M6	Total cost	Demographic Number of visits Drug cost Medical use cost, etc.	R^2^ MAE
Park et al. (2019) [12]	Logistic regression Random forest ANN	Binary (Top 10%)	Demographic Diagnosis Health check-up Medical cost Medical Utilization Variables	Cost-capture AUROC
Itsuki et al. (2020) [13]	Random forest Gradient Boosting Decision tree Lasso regression	Binary (Top 10%)	Demographic Diagnosis Health check-up Medical cost Medical Utilization Variables	C-statistics ROC Variable importance

**Table 2 ijerph-19-13672-t002:** Composition of the analytics dataset.

Input Variable (Current Year)	Total Number (*n*)	High-Cost Medical Expenses in the Next Year (*n*)	Amount of Medical Expenses Based on High Cost in the Current Year (KRW, ₩)	Target Variable (Next Year)
2010	125,349	12,535	₩1,946,262	2011
2011	129,587	12,959	₩2,158,720	2012
2012	124,302	12,431	₩2,056,234	2013
2013	119,896	11,990	₩2,302,485	2014
2014	124,032	12,404	₩2,323,331	2015
2015	133,330	13,333	₩2,641,253	2016
2016	128,872	12,888	₩2,859,968	2017
2017	137,185	13,719	₩3,249,848	2018

**Table 3 ijerph-19-13672-t003:** Training data set and test set descriptive statistics according to data preprocessing methods.

Variables	Training Set (Raw)	Training Set (Down)	Training Set (ROSE)	Test Set
users, N (%)	818,042	162,780	818,042	204,511
male sex, N (%)	464,856 (56%)	88,519 (54%)	445,632 (54%)	118,781 (58%)
AGE, mean	60.39	62.22	62.22	63.84
Disability severity_YES, N (%)	24,255 (3.0%)	7018 (4.3%)	35,311 (4.3%)	23,109 (11.3%)
Disability_YES, N (%)	23,691 (2.9%)	6829 (4.2%)	34,248 (4.2%)	23,109 (11.3%)
BMI, mean	24.10	24.14	24.14	24.27
WAIST, mean	82.26	82.62	82.65	83.02
High blood pressure, mean	124.8	125.4	125.47	125.7
Low blood pressure, mean	77.13	77.07	77.10	76.67
fasting blood sugar, mean	102.8	103.9	103.9	105.3
HDL cholesterol, mean	53.74	53.44	53.45	54.5
LDL cholesterol, mean	116.8	115.4	115.3	113.1
Total cholesterol, mean	196.6	194.8	194.9	193.1
hemoglobin, mean	14.05	13.88	13.88	14.15
proteinuria_YES, N (%)	20,216 (2.5%)	5265 (3.2%)	26,277 (3.2%)	6040 (3%)
glutamic oxaloacetic transaminase, mean	26.49	26.91	26.91	27.29
glutamic pyruvic transaminase, mean	24.96	25.03	25.01	25.19
γ-glutamyl transpeptidase, mean	36.82	38.14	38.17	36.43
TRIGYCERIDE, mean	133.3	132.9	133.1	131
CREATININE, mean	0.967	0.984	0.983	0.893
(Your) Stroke history_YES, N (%)	12,154 (1.5%)	3351 (2.1%)	17,290 (2.1%)	3878 (1.9%)
(Your) history of heart disease_YES, N (%)	34,539 (4.2%)	9531 (5.9%)	48,161 (5.9%)	11,000 (5.4%)
(Your) history of high blood pressure _YES, N (%)	264,214 (32.3%)	58,839 (36.1%)	296,597 (36.3%)	74,393 (36.4%)
(Your) history of diabetes_YES, N (%)	90,424 (11.1%)	22,395 (15.0%)	113,198 (14.9%)	28,152 (13.8%)
(Your) history of dyslipidemia_YES, N (%)	51,475 (6.3%)	10,823 (6.6%)	54,643 (6.7%)	23,041 (11.3%)
(Your) history of tuberculosis_YES, N (%)	14,086 (1.7%)	2,944 (1.8%)	14,478 (1.8%)	3458 (1.7%)
(Your) history of cancer_YES, N (%)	91,048 (11.1%)	24,357 (15%)	122,157 (14.9%)	29,803 (14.6%)
(Family) Stroke history_YES, N (%)	73,227 (9%)	14,062 (8.6%)	70,308 (8.6%)	17,813 (8.7%)
(Family) history of heart disease_YES, N (%)	31,342 (3.8%)	5984 (3.7%)	30,018 (3.7%)	8248 (4%)
(Family) history of high blood pressure_YES, N (%)	113,830 (13.9%)	22,031 (13.5%)	110,618 (13.5%)	28,241 (13.8%)
(Family) history of diabetes_YES, N (%)	75,642 (9.2%)	14,775 (9.1%)	74,126 (9.1%)	34,096 (16.7%)
(Family) history of cancer_YES, N (%)	123,952 (15.2%)	23,930 (14.7%)	120,001 (14.7%)	19,575 (9.6%)
smoking status, N (%)	(1) Non-smoke: 504,878 (61.7%) (2) Past smokers (now quit): 190,213 (23.3%) (3) Smoker: 122,951 (15%)	(1) Non-smoke: 103,490 (63.6%) (2) Past smokers (now quit): 36,198 (22.2%) (3) Smoker: 23,092 (14.2%)	(1) Non-smoke: 520,159 (63.6%) (2) Past smokers (now quit): 181,763 (22.2%) (3) Smoker: 116,120(14.2%)	(1) Non-smoke: 125,300 (61.3%) (2) Past smokers (now quit): 53,706 (26.3%) (3) Smoker: 25,505(12.5%)
Drinking habits, mean	2.14	2.06	2.06	0.88
Number of vigorous exercise for more than 20 min per week, mean	2.39	2.31	2.31	1.27
Number of moderate-intensity exercise for 30 min or more per week, mean	3.71	3.65	3.66	3.10
Number of walking for 30 min or more per week, mean	2.6	2.51	2.52	1.57
Pre_ratio10 high, N (%)	81,704 (10%)	33,601 (20%)	168,719 (20%)	20,555 (10%)
CCI_INDEX, mean	1.38	1.97	1.96	1.47
Number of main-disease diagnosis in the current year, mean	22.2	31.3	31.4	23.8
Number of sub-disease diagnosis in the current year, mean	16.1	23.0	23.0	18.2
Current Expenditure_Top10%, N (%)	81,390 (10%)	81,390 (50%)	408,505 (50%)	20,869 (10%)

**Table 4 ijerph-19-13672-t004:** Description used dataset variables by category.

Variable Category	Variable	Number of Variables
Social qualification	Gender, Age, Disability type, Disability severity, place of residence, Insurance class, Income quantile	7
Health check-up	Body measurement index (BMI, Waist, blood pressure)	32
	Laboratory results	
	(fasting blood sugar, cholesterol, hemoglobin, etc.) Questionnaire, Life style	
Cost	Expenditure by KCD disease group	22
	Top 10% of total medical expenses for the year	
Non-cost	Number of visits by KCD disease group	24
	Number of Diagnosis of Major Diseases	
	Number of Diagnosis of Minor Diseases	
	Charlson Comorbidity Index (CCI) index	

**Table 5 ijerph-19-13672-t005:** Results of prediction model performance evaluation of the entire experimental dataset.

Model	Dataset 1		Dataset 2		Dataset 3		Dataset 4	
	Demographic (7)		Demographic + Health Check-Up (39)		Demographic + Health Check-Up + Cost (61)		Demographic + Health Check-Up + Cost + Non-Cost (85)	
	F1-Score	AUC	F1-Score	AUC	F1-Score	AUC	F1-Score	AUC
Raw-LR	0.237	0.668	0.054	0.692	0.268	0.751	0.299	0.767
Raw-RF	0.019	0.581	0.241	0.689	0.268	0.767	0.274	0.768
Raw-XGBoost	0.029	0.658	0.046	0.695	0.000	0.674	0.000	0.667
Down-LR	0.238	0.668	0.247	0.692	0.306	0.760	0.323	0.769
Down-RF	0.243	0.652	0.241	0.688	0.293	0.765	0.304	0.766
Down-XGBoost	0.240	0.653	0.257	0.692	0.319	0.769	0.323	0.771
Rose-LR	0.238	0.668	0.249	0.691	0.306	0.742	0.332	0.762
Rose-RF	0.242	0.641	0.279	0.684	0.000	0.610	0.005	0.600
Rose-XGBoost	0.237	0.646	0.256	0.683	0.319	0.659	0.002	0.500

**Table 6 ijerph-19-13672-t006:** Results of prediction model performance evaluation of variable type (down-XGBoost).

Model	Demographic (7)		Demographic + Health Check-Up (39)		Demographic + Cost (29)		Demographic + Non-Cost (31)	
	F1 Score	AUC	F1 Score	AUC	F1 Score	AUC	F1 Score	AUC
Down- XGBoost	0.240	0.653	0.257	0.692	0.305	0.764	0.304	0.754

**Table 7 ijerph-19-13672-t007:** Top 30 Variable importance (Down-sampling dataset 4) of XGBoost and Random Forest.

	XGBoost		Random Forest	
Variable Category	Variable	Gain	Variable	Mean Decrease Accuracy
Social qualification	AGE	0.048	AGE	102.2
Health check-up	BMI	0.020	HMG	40.0
	HMG	0.018	TOT_CHOLE	31.9
	GAMMA_GTP	0.018	WAIST	31.4
	BLDS	0.016	BMI	29.4
	TRIGLYCERIDE	0.016	LDL_CHOLE	27.4
	TOT_CHOLE	0.014	SGPT_ALT	24.9
	LDL_CHOLE	0.014		
	SGOT_AST	0.012		
	WAIST	0.012		
	BP_HIGH	0.012		
	SGPT_ALT	0.011		
	BP_LWST	0.009		
	CREATININE	0.009		
cost	Pre_ratio10high	0.209	pre_ratio10	78.3
	G13M	0.029	GR13M	72.5
	GR2CD	0.016	GR2CD	59.6
	GR11K	0.014	GR10J	47.9
	GR7H	0.014	GR5F	44.1
	GR10J	0.013	GR9I	40.9
	GR14N	0.012	GR19ST	40.0
	GR19ST	0.012	GR14N	39.6
	GR18R	0.009	GR7H	37.4
	GR5F	0.009	GR11K	35.9
	GR12L	0.009	GR6G	34.1
	GR4E	0.009	GR4E	33.4
	GR6G	0.008	GR18R	31.1
			GR21Z	30.2
non-cost	Y_MC	0.164	Y_MC	63.6
	Y_SC	0.043	CCI_INDEX	56.1
	CCI_INDEX	0.022	Y_SC	53.5
			GR13M_C	48.2
			GR2CD_C	39.4
			GR10J_C	30.3
			GR11K_C	27.0
			GR5F_C	25.0
			GR7H_C	24.5

## Data Availability

Restrictions apply to the availability of these data. Data was obtained from National Health Insurance Sharing Service (NHISS) and are available from [https://nhiss.nhis.or.kr/bd/ab/bdaba021eng.do (accessed on 15 August 2022)] with the permission of NHISS.

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
