# Peer review of "Development and Evaluation of Machine Learning-Based High-Cost Prediction Model Using Health Check-Up Data by the National Health Insurance Service of Korea"

_ijerph, 2022, doi:10.3390/ijerph192013672_

Round 1

Reviewer 1 Report

This is a timely and much needed study. I like the conceptualisation of the project: using the existing national insurance claiming data, clinical data and health check data to predict high cost patients. However, the manuscript is poorly written. The language is not clear. The structure is poor. For example, the authors appear to forget the aim and objectives coming to the conclusion section. They continue to present findings, which should be presented in the results section. The paper suffer from sever language deficits. I have to guess the meaning of the sentences most of the time. Therefore, as it is written now, the paper is not up to the level of clarity for publication. I provide the following examples that I would like clarification. The unclear sentences full of the paper, making it worthless to pinpoint the exact deficits. In the end, it is the responsibility of the authors to present to the international audience a paper that is written in clear English language.

Figure 1. Medical expenses? What is the unit? Per person or the whole nation?

Table 2. What is the definition of high-cost medical expenses? What is the unit of analysis for each variable in the table?

Other than that, I trust the methodology and the findings.

Reviewer 2 Report

1. What is the novelties in this ''Development and evaluation of machine learning-based high cost prediction'' model. Please more explanation about it in the abstract, introduction, and literature review.

2. What is the gaps based on the investigated papers in the literature review?  Contributions are unclear. This should be made clear from the very beginning of the paper till its end.

3. How authors use of this ''big data for health check-up'' in this paper? please authors should explain clearly in the manuscript.

4. There is no scope for future research. A clear direction for future research is required.

5.Authors should present introduction section better based on novelties of their work.

6. Please check all text to remove grammatical errors.

7. Please report only relevant information of your work and novelty in conclusion section.

8. Please add this word in the Abstract ''KCD(Korea Classification Disease)''. Not in key words. 

9. The literature review is brief. Some of the included papers could be briefly described. Also, a general overview of the topic could also be included. For instance, the following could be added: *Hybrid meta-heuristic algorithms for a supply chain network considering different carbon emission regulations using big data characteristics. Soft Computing, 25(11), 7527-7557.

Reviewer 3 Report

An empirical study conducted by analyzing more than one million actual medical claim data and clinical data using public health and medical big data provided by the Korean government is presented in this paper. Through the study, it was confirmed that musculoskeletal disease (M) and respiratory system disease (J), which are the most frequently treated diseases as major disease groups for high cost prediction of major disease groups in KCD, are disease groups that affect high cost prediction in the future. It was confirmed that cancer diseases are a group of diseases related to the prediction of high future medical expenses.

 It will be good to clear if really "big data" had been processed. One million static structured data instances are not big data. The basic definition of this concept is that "it is a large volume of semi-structured dynamic data that cannot be processed with conventional systems and algorithms"!

The authors of the article use conventional algorithms for processing previously prepared tables with well-structured static data. This means that they work with a large amount of data instances, but not with Big Data.
